# Periphery-Fused Chiral A_2_B-Type Subporphyrin

**DOI:** 10.3390/molecules26041140

**Published:** 2021-02-20

**Authors:** Shoma Hirokawa, Nagao Kobayashi, Soji Shimizu

**Affiliations:** 1Department of Chemistry, Graduate School of Science, Tohoku University, Sendai 980-8578, Japan; shoma-1212.vassago.9@i.softbank.jp; 2Faculty of Textile Science and Technology, Shinshu University, Ueda 386-8567, Japan; 3Department of Chemistry and Biochemistry, Graduate School of Engineering and Center for Molecular Systems (CMS), Kyushu University, Fukuoka 819-0395, Japan

**Keywords:** subporphyrin, chirality, circular dichroism, magnetic circular dichroism

## Abstract

Despite significant interest, the chiroptical properties of subporphyrins have rarely been investigated because chiral subporphyrins are elusive. Here, inherently chiral subporphyrins are elaborated by forming a fused pyran ring at the periphery of an A_2_B-type *meso*-aryl-substituted subporphyrin. Their circular dichroism (CD) properties are largely affected by the peripheral substituents and the dihedral angles between the *meso*-aryl substituents and the subporphyrin core: the β-perbromo subporphyrin with an orthogonal arrangement of the *meso*-phenyl substituents to the subporphyrin core exhibits weak CD signals corresponding to the Q bands, whereas the unsubstituted species with smaller dihedral angles shows relatively intense CD signals. A detailed structure–property relationship of these chiral subporphyrins was elucidated by time-dependent (TD) DFT calculations. This study reveals that the CD properties of chiral subporphyrins can be controlled by peripheral substitution and *meso*-aryl substituents.

## 1. Introduction

Subporphyrin, exclusively known as its boron complex form, is a ring-contracted analog of porphyrin comprising three pyrrole rings (Figure 1a) [1,2]. In contrast to the first synthesis of its *meso*-nitrogen counterpart, subphthalocyanine (Figure 1b), in the early 1970s [3], subporphyrin was elusive until the recent, seminal syntheses of tribenzosubporphine by Osuka et al. in 2006 [4] and *meso*-aryl-substituted subporphyrins by Kobayashi and Osuka in 2007 [5,6,7]. As a result of its bowl-shaped structure arising from the tetrahedral coordination of the central boron atom, subporphyrin has been drawing attention as a curved π-conjugated system [8], as with corannulene [9,10] and sumanene [11]. Among various properties expected for this unique curved structure such as concave-convex π,π-interactions [12] and bowl-to-bowl inversion, bowl chirality and chiroptical properties such as circular dichroism (CD) and circularly polarized luminescence (CPL) are of significant interest in terms of their potential application in supramolecular chemistry and optoelectronics.

Recently, Osuka et al. reported the synthesis and optical resolution of so-called ABC-type subporphyrin containing three different kinds of *meso*-aryl substituent [13]. However, the CD signals of the enantiomers were extremely weak because the chirality induced by the *meso*-aryl substituents has only a minor effect on the chiroptical properties of the subporphyrin core. To enhance the chiroptical properties, it is, therefore, essential to introduce chirality to the π-conjugated system of subporphyrin. During our studies on subphthalocyanine, we found that asymmetrization of the original *C*_3v_ symmetry to the lower *C*_3_ or *C*_1_ symmetry by annulation of aromatic ring units, which led to the inherently chiral structures of 1,2-subnaphthalocyanines, induced intense CD properties (Figure 1c) [14,15]. On the basis of these results, we examine here a peripheral ring-closure reaction between the *meso*-aryl substituent and the pyrrolic β-position of A_2_B-type subporphyrin to generate inherently chiral subporphyrin systems and reveal their structure–chiroptical property relationship.

## 2. Results and Discussion

Periphery-fused chiral A_2_B-type subporphyrins were synthesized as shown in Scheme 1. Achiral A_2_B-type subporphyrin **1** containing two phenyl and one *o*-anisyl substituents was synthesized according to the procedure reported by Osuka et al. [16] After conversion of the methoxy group of the *o*-anisyl substituent to a hydroxy group by a reaction with BBr_3_, bromination of **2** was conducted using bromine in chloroform to provide a perbromo-compound (**3**). Then, intramolecular ether formation of **3** was performed in the presence of potassium carbonate in DMF. A racemic mixture of **4** bearing a fused pyran ring was obtained in 14% overall yield from **1**. The remaining peripheral bromo substituents were removed by a two-step reaction, lithiation with *n*-butyl lithium and subsequent hydrolysis by quenching with water. A trace amount of β-unsubstituted, periphery-fused chiral A_2_B-type subporphyrin **5** was obtained as a racemic mixture. Because **5** was easily racemized at elevated temperatures of around 40 °C, probably due to the bowl-to-bowl inversion process in the axial ligand exchange dynamics (vide infra), **5** was converted to **6** with an axial phenyl ligand by a reaction with phenylmagnesium bromide according to the literature procedure [17]. These pyran-fused subporphyrins are structural analogs of thiopyran-fused subporphyrin reported by Osuka et al. [18] All compounds were characterized by high-resolution mass spectrometry (HR-MS) and ^1^H NMR spectroscopy (Appendix A).

The HR-MS result of **4** indicated that two more C–H bonds were brominated in addition to the five β-pyrrolic C–H bonds. Although the crystal structure of **4** has not yet been obtained, we revealed that the 3,5-positions of the fused phenyl ring were brominated because two singlets were observed at 9.55 and 8.08 ppm in the ^1^H NMR spectrum of **4** at room temperature (Figure 2). This assignment is also convincing considering the electrophilic bromination of these two positions in **3** can be promoted by the *ortho*- and *para*-directing hydroxy group. At room temperature, no proton signal of the *meso*-phenyl substituents was observed except for the *para*-proton signal. In the case of A_3_-type *meso*-phenyl-substituted subporphyrin, one *ortho*-proton signal and one *meta*-proton signal are observed in the ^1^H NMR spectrum at room temperature due to the free rotation of phenyl rings. The absence of *ortho*- and *meta*-proton signals of **4** can be ascribed to the steric hindrance of the β-pyrrolic C–Br bonds, which can slow down the rotation of the phenyl rings to broaden the *ortho*- and *meta*-proton signals. Upon lowering the temperature, the broad signals gradually appeared and became sharp doublets for the *ortho*-protons at 8.22, 8.14, 7.10, and 7.06 ppm and multiplets for the *meta*-protons at 7.75 and 7.44 ppm at −60 °C due to the hindered rotation of the phenyl rings (Figure 2). The splitting of the *ortho*- and *meta*-proton signals reflect different deshielding ring current effects between the concave and convex sides [19]. The activation barrier of **4** for the rotation of the phenyl rings was estimated to be 57 kJ mol^–1^ by the coalescence method using the coalescence temperature of 20 °C and signal separation of 217 Hz for the *meta*-protons. This activation barrier is fairly comparable with those of the A_3_-type β-perbromo-*meso*-aryl-substituted subporphyrins [20].

In contrast to the broad ^1^H NMR spectrum of **4** at room temperature, **5** and **6** exhibit sharp multiplets due to the facile rotation of the phenyl rings (Appendix A). The axial phenyl proton signals of **6** observed in the up-field region indicate successful conversion of the axial ligand from a hydroxy group to a phenyl group by a reaction of **5** with phenylmagnesium bromide.

All the chiral compounds (**4**–**6**) were separated into a pair of enantiomers using HPLC equipped with a preparative chiral column. Here, **Fr1** and **Fr2** in the compound names denote the first and second eluted fractions. Despite the successful chiral resolution, enantiomers of **5** were racemized during removal of the solvent under vacuum around 40 °C. A facile racemization of **5** in solution was also seen in the CD measurements, which revealed that the CD signals of a freshly isolated sample gradually decreased at 40 °C in chloroform (Appendix A). Because **6Fr1** and **6Fr2** with an axial phenyl ligand were not racemized under similar conditions, the racemization of **5** may proceed through a bowl-to-bowl inversion mechanism via a planar subporphyrin cation formed in a S_N_1-type heterolysis of the axial B–OH bond in solution at elevated temperatures (Scheme 2) [21]. Although similar S_N_1-type heterolysis of **4** can occur in solution, thermal racemization was not observed for **4Fr1** and **4Fr2** at a similar temperature range as tested for racemization of **5**. This can be explained in terms of the steric hindrance between the β-pyrrolic C–Br bond and the *ortho* C–H bond of the fused phenyl group, which may prevent **4** from taking a planar cation conformation.

In the absorption spectra of **4** and **6**, the spectral profiles in the Soret band region (350–400 nm) are broadly similar, whereas those in the Q band region (425–550 nm) are slightly different: **4** exhibits a more distinct shoulder absorption at 475 nm compared with the gentle slope of the Q band of **6** (Figure 3). The fluorescence spectrum of **6** shows emission at 558 nm, and the fluorescence quantum yield is 0.16.

In contrast to broadly similar absorption spectra, **4** and **6** exhibit apparently different magnetic circular dichroism (MCD) spectra [22,23], featured by first derivative band shapes corresponding to the Q and Soret band absorptions, with peaks and troughs of 520, 504, 392, and 378 nm for **4** and 527, 501, and 397 nm for **6**, respectively. These MCD signals are assigned as pseudo Faraday *A* terms [24,25]. A pseudo Faraday *A* term is observed when two Faraday *B* terms appear in close energy due to accidental degeneracy of the excited states of molecules with less than threefold symmetry. The MCD sign sequence of **4** is negative-to-positive (520 and 504 nm) and positive-to-negative (392 and 378 nm) in ascending energy for the Q and Soret bands, respectively. **6** exhibits a negative-to-positive sign sequence in the Q band region (527 and 501 nm), and only the negative sign is observed for the Soret band (397 nm).

These kinds of anomalous MCD signs in the Soret band region (the positive-to-negative sign sequence for **4** and the absence of a positive sign for **6**) were also observed for A_3_-type *meso*-aryl-substituted subporphyrins in the previous studies [5,6]. In the case of a series of A_3_-type *meso*-aryl-substituted subporphyrins, going from electron-donating groups (EDGs), such as *p*-anisyl group, to electron-withdrawing groups (EWGs), such as *p*-trifluoromethylphenyl and 3- and 4-pyridyl groups, the MCD sign sequence in the Soret band region changes from a negative-to-positive pattern (i.e., a positive Faraday *A* term) to a positive-to-negative pattern (i.e., a negative Faraday *A* term) in ascending energy. According to Michl’s perimeter model for 4N + 2 π-electron systems [26,27,28], the MCD sign sequences of Faraday *A* and *B* terms are determined by the relative magnitude difference between the energy gap of the HOMO and the HOMO–1 (ΔH) and that of the LUMO and the LUMO+1 (ΔL): a negative-to-positive MCD sign sequence is observed when ΔH > ΔL, whereas the opposite positive-to-negative sign sequence is observed when ΔL > ΔH. Considering the degeneracy of the LUMO in the threefold molecular symmetry of the A_3_-type *meso*-aryl-substituted subporphyrins (i.e., ΔH > ΔL), the negative Faraday *A* term in the Soret band region observed for the subporphyrins with EWGs at the *meso*-positions is unusual. Ceulemans and Kobayashi ascribed this MCD sign anomaly to quenching of the angular momentum of the excited electron relative to that of the positive charge left in the occupied molecular orbitals (MOs) due to the electron accepting nature of the EWGs at the *meso*-positions [29]. This quenching significantly affects the sign and strength of the magnetic moment of the Soret band transitions compared with those of the Q band transitions. Therefore, the type of *meso*-aryl substituents can alter the MCD sign sequence of the A_3_-type *meso*-aryl-substituted subporphyrins in the Soret band region.

The MCD spectral profile of **4** resembles that of electron-withdrawing 4-pyridyl-substituted A_3_-type subporphyrin, whereas **6** exhibits a similar MCD spectral profile to electron-donating *p*-anisyl-substituted A_3_-type subporphyrin. As detailed below, the observed MCD sign anomalies of **4** and **6** can be explained in a similar manner to the case of the A_3_-type *meso*-aryl-substituted subporphyrins. The partial frontier MOs and time-dependent DFT (TDDFT) calculations, for which model structures with an axial chloro ligand (**4’** and **6’**) were used for simplicity, revealed the major contribution of the transitions between four frontier MOs (HOMO–1, HOMO, LUMO, and LUMO+1) to the Soret and Q bands, indicating that **4** and **6** retain the electronic structure of subporphyrin as a core chromophore structure despite the presence of the fused pyran ring (Figure 4, Table 1, Appendix A).

In the MO diagram, the four frontier MOs of **4’** are stabilized compared with those of **6’** due to the electron-withdrawing inductive effect of the peripheral bromo groups. On the other hand, the antibonding MO interactions of the bromo groups to the subporphyrin core destabilize the HOMO–1 to decrease ΔH on going from **6’** (ΔH = 0.50 eV) to **4’** (ΔH = 0.37 eV). Due to the absence of threefold symmetry, the degeneracy of the LUMO is lifted, giving a ΔL of 0.21 eV for **4’** and 0.15 eV for **6’**. The ΔH > ΔL relationship for both **4’** and **6’** agrees with the observed positive pseudo Faraday *A* term corresponding to the Q band. The small MCD intensities of **4** in the Q band region are ascribed to the enhanced forbidden nature of the Q bands by configurational interactions of the Soret and Q band transitions due to the similar magnitudes of ΔH and ΔL. The negative MCD sign of **6** in the Soret band region also agrees with the ΔH > ΔL relationship despite the absence of the positive MCD sign corresponding to the Soret band in the higher energy, which is probably canceled by overlapping with intense negative MCD signals in the lower energy region. The negative pseudo Faraday *A* term of **4** in the Soret band region is discrepant from the MCD sign prediction based on the ΔH > ΔL relationship. As with the A_3_-type *meso*-aryl-substituted subporphyrins bearing EWGs, the magnetic moment arising from the excited electron can be quenched by the electron-withdrawing bromo groups at the β-positions. Unlike the conventional *meso*-aryl-substituted subporphyrins, *meso*-aryl groups may have a minor effect on the electronic structure of **4** due to the nearly orthogonal arrangement of the *meso*-aryl groups and the subporphyrin core. The resulting dominant contribution of the magnetic moment arising from the positive charge induces the observed negative pseudo Faraday *A* term of **4** in the Soret band region.

The CD spectra of the first and second eluted fractions of **4** (**4Fr1** and **4Fr2**) and **6** (**6Fr1** and **6Fr2**) are mirror images of each other (Figure 3). The CD spectral profiles of **4** and **6** are similar in the Soret band region, exhibiting a first derivative band shape corresponding to the Soret band absorption. In contrast, the spectral profiles and intensities are different in the Q band region: **4** exhibits a derivative band shape (512 and 494 nm) corresponding to the Q band at 510 nm, whereas relatively intense CD signals with a similar spectral profile to the Q band absorption irrespective of its sign are observed for **6**.

The TDDFT calculations at the B3LYP/6-311G(d,p) level reproduces well the observed CD spectra (Appendix A). On the basis of the calculated CD signals, the absolute structures of the first and second eluted fractions are assigned to ***M*** and ***P*** isomers, respectively (Figure 5). ***P*** and ***M*** isomers represent structures with the fused pyran ring on the right and left side, respectively, when viewed from the convex side. To give an insight into the different CD signals of **4** and **6** in the Q band region, transition electric (TEDM (***μ***)) and transition magnetic (TMDM (***m***)) dipole moments, angles of TEDM and TMDM ($\theta_{\mu \cdot m}$), and rotatory strengths ($R\;=\;\left|\mu \right|\cdot \left|m\right|\cdot \cos\theta_{\mu \cdot m}$) were analyzed (Table 2). Because of the similar magnitudes of both TEDM and TMDM of the first two transitions, the larger CD intensity of **6** compared with **4** can be ascribed to slight deviations of $\theta_{\mu \cdot m}$ from the orthogonal angle (Table 2 and Appendix A). Considering the same core chromophore structures for **4** and **6**, this analysis reveals that peripheral substitution and dihedral angles between the *meso*-phenyl substituents and the subporphyrin core play an important role in determining the CD intensities. To estimate the perturbation of the *meso*-phenyl substituents, the CD properties of a model structure of **6** (**6’*P*-90**) with orthogonal arrangement of the *meso*-phenyl substituents and the subporphyrin core were calculated (Table 2). Despite the wrong positive CD sign predicted for the fourth transition (**6’*P*-90**: *R* = 18.8 (λ = 352 nm)), the calculated CD of **6*P*-90** also predicts large rotatory strengths for the Q band transitions, indicating that not only the dihedral angles of the *meso*-phenyl substituents, but also the electronic effects of the peripheral substituents affect the rotatory strengths.

## 3. Materials and Methods

### 3.1. General Procedure

High-resolution mass spectrometry (HR-MS) was performed on a Bruker Daltonics solariX 9.4T spectrometer (FT-ICR (Fourier transform ion cyclotron resonance) mode). ^1^H NMR spectra were recorded on a Bruker AVANCE 500 (operating at 500.133 MHz) spectrometer or on a JEOL ECA-700 (operating at 700.125 MHz) using the residual solvent as the internal reference for ^1^H (δ = 7.26 ppm for CDCl_3_). Electronic absorption spectra were recorded on a JASCO V-570 spectrophotometer. Fluorescence spectra and absolute fluorescence quantum yields were measured using a Hamamatsu Photonics A10104-01 calibrated integrating sphere system. CD and MCD spectra were recorded on a JASCO J-725 spectropolarimeter. In the MCD measurements, an electromagnet, which produces magnetic fields of up to 1.03 T (1T = 1 tesla) with both parallel and antiparallel fields, was placed in a cell compartment. The magnitudes were expressed in terms of molar ellipticity ([θ]/deg dm^3^ mol^–1^ cm^–1^) for CD spectra and molar ellipticity per tesla ([θ]_M_/deg dm^3^ mol^–1^ cm^–1^ T^–1^) for MCD spectra. Preparative separations were performed by silica gel column chromatography (Silica gel 60, Merck), alumina gel column chromatography (Wako), gel permeation chromatography (GPC) (Bio-Beads S-X1) and recycling preparative GPC-HPLC (JAI LC-9210 NEXT with preparative JAIGEL-2.5H and 3H columns). Optical resolution was performed on an HPLC (SHIMADZU LC-6AD with a SHIMADZU SPD-M10AVP detector) equipped with a chiral column (DAICEL CHIRALPAK IA). All reagents and solvents were of commercial reagent grade and were used without further purification except where noted.

### 3.2. Theoretical Calculation Details

The Gaussian 16 software package [30] was used to conduct DFT and TDDFT calculations using the B3LYP functional [31,32] with a 6-311G(d,p) basis set [33]. Structural optimization was performed on model structures in which the axial ligand was replaced with a chloro group for simplicity.

### 3.3. Synthetic Procedures

#### 3.3.1. (5-(2-Hydroxyphenyl)-10,15-diphenylsubporphyrinato)boron(III) **2**

5-(*o*-Anisyl)-10,15-subporphyrin with a methoxy axial ligand (**1**) was synthesized according to the literature procedure [16]. To a dichloromethane solution (50 mL) of **1** (106 mg, 0.20 mmol) was added a 1 M dichloromethane solution of boron tribromide (1.0 mL, 5.0 eq). Then, the reaction mixture was stirred at room temperature for three hours. The reaction mixture was washed with brine, and the solvent was removed. The obtained mixture was used for the synthesis of **3** without further purification. 

#### 3.3.2. (2,3,7,8,12,13-Hexabromo-5-(3,5-dibromo-2-hydroxyphenyl)-10,15-diphenylsubporphyrinato)boron(III) **3**

An excess amount of bromine (0.3 mL, 5.8 mmol) was added to a chloroform solution (50 mL) of the mixture containing **2**, which was obtained from the above-mentioned reaction, and the mixture was stirred at 60 °C. After confirming the consumption of **2**, the mixture was poured into an aqueous solution of sodium thiosulfate. The organic layer was extracted, and the solvent was removed. The mixture was used for the synthesis of **4** without further purification.

#### 3.3.3. Axially-Hydroxy-Substituted Pyran-Fused Perbromo-Subporphyrin **4**

An excess amount of potassium carbonate (3.0 g, 22 mmol) was added to a DMF solution (40 mL) of the mixture containing **3**, which was obtained from the above-mentioned reaction, and the resultant mixture was stirred at 80 °C for three hours. The mixture was poured into brine. The organic layer was extracted, and the solvent was removed. The residue was purified by silica gel column chromatography (eluent: dichloromethane) and GPC-HPLC (eluent: chloroform). The crude sample was recrystallized from dichloromethane and hexane to provide **4** (30 mg, 53 μmol) as an orange solid in 14% yield from **1**.

HR-MALDI-FT-ICR-MS: *m/z* calcd. for C_33_H_13_BBr_7_N_3_O_2_: 1046.5379, [M]^+^: 1046.5377. ^1^H NMR (CDCl_3_, 500 MHz, 298 K): δ [ppm] = 9.55 (d, *J* = 2.3 Hz, 1H; pyran-fused phenyl), 8.08 (d, *J* = 2.3 Hz, 1H; pyran-fused phenyl), 7.68 (t, *J* = 7.3 Hz, 2H; *para*-phenyl), 7.61 (brs), 0.07 (s, 1H; axial-OH); ^1^H NMR (CDCl_3_, 700 MHz, 213 K): δ [ppm] = 9.44 (d, *J* = 2.3 Hz, 1H; pyran-fused phenyl), 8.22 (d, *J* = 7.3 Hz, 1H, *ortho*-phenyl), 8.14 (d, *J* = 7.3 Hz, 1H; *ortho*-phenyl), 8.06 (d, *J* = 2.3 Hz, 1H, pyran-fused phenyl), 7.75 (m, 2H; *meta*-phenyl), 7.67 (t, *J* = 7.3 Hz, 2H; *para*-phenyl), 7.44 (m, 2H; *meta*-phenyl), 7.10 (d, *J* = 7.3 Hz, 1H; *ortho*-phenyl), 7.06 (d, *J* = 7.3 Hz, 1H; *ortho*-phenyl), 0.07 (s, 1H; axial-OH). UV/vis (CH_2_Cl_2_): λ_max_ [nm] (ε [mol^–1^dm^3^cm^–1^]) = 510 (19000), 475 (12000), 388 (110000).

#### 3.3.4. Axially-Hydroxy-Substituted Pyran-Fused Subporphyrin **5**

To a THF solution (40 mL) of **4** (16 mg, 15 μmol) was added an excess amount of 1.6 M hexane solution of *n*-butyl lithium (5.0 mL, 8 mmol) at –78 °C, and the mixture was stirred at –78 °C for one hour. The reaction was quenched with brine, and the solvent was removed under vacuum. The mixture was purified by silica gel column chromatography (eluent: dichloromethane/methanol = 40:1 (*v:v*)) to provide a trace amount of **5** as a yellow solid.

HR-MALDI-FT-ICR-MS: *m/z* calcd. for C_33_H_19_BN_3_O: 483.1652, [M–OH]^+^: 483.1652. ^1^H NMR (CDCl_3_, 500 MHz, 298 K): δ [ppm] = 9.00 (m, 1H), 8.32 (m, 1H), 8.10 (m, 1H), 8.07–8.01 (m, 6H), 7.73 (m, 1H), 7.69 (m, 4H), 7.62 (m, 4H), 7.35 (s, 1H). The axial hydroxy proton signal was not observed due to the overlap of residual solvent peaks. Proton signals could not be assigned because the obtained trace amount of the sample was insufficient for further characterization. UV/vis (CH_2_Cl_2_): λ_max_ [nm] = 506, 381.

#### 3.3.5. Axially-Phenyl-Substituted Pyran-Fused Subporphyrin **6**

To a 1,4-dioxane solution (6.0 mL) of **5** (~1.0 mg, ~2.0 μmol) was added an excess amount of 1.1 M THF solution of phenylmagnesium bromide (4.0 mL, 4.4 mmol), and the mixture was refluxed for three hours. The reaction mixture was poured into an aqueous solution of ammonium chloride, and the organic layer was extracted. After removal of solvent, the residue was purified by alumina gel column chromatography (eluent: dichloromethane/ethyl acetate = 2:1 (*v:v*)) and GPC HPLC (eluent: chloroform). Finally, **6** was isolated by dissolving with hexane, and a trace amount of **6** was obtained as a red solid.

HR-MALDI-FT-ICR-MS: *m/z* calcd. for C_33_H_19_BN_3_O: 483.1652, [M–Ph]^+^: 483.1652. ^1^H NMR (CDCl_3_, 500 MHz, 298 K): δ [ppm] = 9.00 (m, 1H; pyran-fused phenyl), 8.33 (d, *J* = 4.5 Hz, 1H; β-pyrrolic), 8.12 (d, *J* = 4.5 Hz, 1H; β-pyrrolic), 8.09–8.02 (m, 6H; *ortho*-phenyl and β-pyrrolic), 7.73 (m, 1H, pyran-fused phenyl), 7.68 (m, 4H; *meta*-phenyl), 7.60 (m, 4H; *para*-phenyl and pyran-fused phenyl), 7.39 (s, 1H; β-pyrrolic), 6.52 (m, 1H; axial-*para*-phenyl), 6.40 (dd, *J*_1_
*= J*_2_ = 6.5 Hz, 2H; axial-*meta*-phenyl), 4.89 (d, *J* = 6.5 Hz, 2H; axial-*ortho*-phenyl). Proton signals were tentatively assigned by the theoretical ^1^H NMR calculations (Appendix A) and by comparing the spectrum with that of the thiopyran-fused analog [18]. Two dimensional NMR measurements for further assignment could not be performed due to the trace amount of the sample. UV/vis (CH_2_Cl_2_): λ_max_ [nm] = 519, 389.

## 4. Conclusions

In summary, inherently chiral subporphyrins containing a fused pyran ring were synthesized by peripheral bromination of A_2_B-type *meso*-aryl-substituted subporphyrin, intramolecular ether formation, and debromination by lithiation and subsequent hydrolysis. In the absence of peripheral steric hindrance, the axial phenyl ligand in **6** was indispensable for maintaining the chiral structures because the axially hydroxy-substituted compound (**5**) exhibited a facile racemization via a planar borenium cation formed by dissociation of the B–O bond. The observed CD properties and calculated optical properties indicate that not only the peripheral substitution, but also the dihedral angles between the *meso*-aryl substituents and the subporphyrin core play a crucial role in determining their signs and intensities. Considering the fluorescence emission of subporphyrin in the visible region, the knowledge acquired in this study will be of benefit in creating efficient circularly polarized luminescent materials. Further investigations along this direction are being intensively investigated in our laboratory.

## Data Availability

Data is contained within the article and Appendix A.

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
