# Peer review of "Periphery-Fused Chiral A2B-Type Subporphyrin"

_molecules, 2021, doi:10.3390/molecules26041140_

Round 1
Reviewer 1 Report
This manuscript entitled: Periphery-Fused Chiral A2B-Type Subporphyrin describes the preparation of apparently new subporfirins. After a short introductory part, target substances were prepared using standard procedures. Then the prepared substances were characterized by NMR study, UV-VIS, CD, fluorescence spectra and DFT calculations. Due to the application potential of subporphyrins, I consider the presented results to be an important extension in the field of chemistry and physical properties of subporphyrins. Nevertheless, I find incomplete information in the article, the addition of which will improve the quality of work:
The characterization of the prepared substances is not complete and 13C NMR spectra are missing. I also don't understand why the X-ray structures of 4.5 or 6 were not measured. Did you try to crystallize 4,5,6?
The description of the performed experiments is also incomplete and its quality needs to be improved. In this article (Chem. Eur. J. 2014, 20, 16194 - 16202) you will find an example of what a quality experimental part should look like.
Scheme 1 does not show isolated yields. Please complete them.
In Scheme 1, the substituent X for substances 2,3 is given, but the definition of X is missing
Page 9, line 292: There is „(CDCl3, 700 MHz, 213 K)“ should be „1H NMR (CDCl3, 700 MHz, 213 K)“
Page 9, line 276: There is " Hexabormo" should be "Hexabromo"
Author Response
1) The characterization of the prepared substances is not complete and 13C NMR spectra are missing. I also don't understand why the X-ray structures of 4, 5 or 6 were not measured. Did you try to crystallize 4, 5, 6?
Response: Because of the limited amounts of obtained samples (4, 5, and 6), we were unable to measure the 13C NMR spectra. Because of the same reason, we have not been able to obtain suitable single crystals for X-ray diffraction analysis.
2) The description of the performed experiments is also incomplete and its quality needs to be improved. In this article (Chem. Eur. J. 2014, 20, 16194–16202) you will find an example of what a quality experimental part should look like.
Response: The experimental section was revised and described in detail as much as possible.
Scheme 1 does not show isolated yields. Please complete them.
Response: I added the isolated yields in Scheme 1. We have not determined the yields of 2 and 3 because these compounds were used for the following reaction without isolation.
In Scheme 1, the substituent X for substances 2,3 is given, but the definition of X is missing.
Response: The axial ligands of 2 and 3 were not identified because these compounds were not isolated. In response to this comment, I added a sentence, "The axial ligands of 2 and 3 denoted by X were not identified.", to the scheme caption.
Page 9, line 292: There is „(CDCl3, 700 MHz, 213 K)“ should be „ H NMR (CDCl3, 700 MHz, 213 K)“
Page 9, line 276: There is " Hexabormo" should be "Hexabromo"
Response: I made corrections according to these comments.
Reviewer 2 Report
In this manuscript the authors report on their synthesis and characterization of a meso-aryl substituted subporphyrin having a pyran ring fused to its periphery in an effort to be able to test the circular dichroism (CD) of the separated enantiomers. Subporphyrins have only been successfully synthesized a decade and a half ago and their meso-substition by three different aryl groups has been found to exhibit only very weak CD in 2014. Hence the effort to probe an intrinsically optically active chromophore of this familly of compounds whose domed structures have already attracted attention for their potential applications in supramolecular chemistry despite their relative novelty.
The synthesis calls for the bromination of the three pyrrole groups and leads to a double bromination of the anisol meso-substituant before the formation of an ether bond closes the intended pyran ring. This yields one of the species whose enantiomers are being tested for their optical activity, while the target compound is obtained after removal of all the bromo substituants. The most striking result of this study is the strength of the CD and magnetic CD response of the Q-band of the final compound compared to the corresponding rather weak signals in the brominated parent enantiomers. Through DFT and TD-DFT calculations at an appropriate level of theory, the authors conclude that it cannot be the fused part by itself that explains the strong activity of the target chromophore because it is common to both the brominated and unbrominated species.
Through DFT calculations carried out by fixing the dihedral angle of the two meso-phenyl groups they come to the conclusion that since low dihedral angles lead to stronger calculated Q-band activity, it is the free rotation of the two meso-aryl substituants which is essential.
The paper is written up clearly and appropriately organized. The background and existing literature is adequately delineated. The procedures, characterization methods, spectral measurements and DFT calculations are sufficiently documented and appear sound. The author's conclusions seem warranted and are worthy of publication.
The only thing that may need to be modified before publication is in the drawings in Scheme 2: the two structures on the right and which are indicated as identical are not! The fused pyran/phenyl rings shown in the third structure from the left need to be positioned just as in first two. Only them will the third structure represent the same enantiomer as the fourth!
Author Response
The only thing that may need to be modified before publication is in the drawings in Scheme 2: the two structures on the right and which are indicated as identical are not! The fused pyran/phenyl rings shown in the third structure from the left need to be positioned just as in first two. Only them will the third structure represent the same two. Only them will the third structure represent the same enantiomer as the fourth!
Response: Thank you for the suggestion. I revised the Scheme 2, and the two structures on the right are the same enantiomer.
Round 2
Reviewer 1 Report
The authors responded adequately to the comments.